# Finite-Time Synchronization of Markovian Jumping Complex Networks with Non-Identical Nodes and Impulsive Effects

**DOI:** 10.3390/e21080779

**Published:** 2019-08-08

**Authors:** Tao Chen, Shiguo Peng, Zhenhua Zhang

**Affiliations:** School of Automation, Guangdong University of Technology, Guangzhou 510006, China

**Keywords:** finite-time synchronization, Markovian jumping, impulsive effects, non-identical node, complex networks

## Abstract

In this paper, we investigate the finite-time synchronization problem for a class of Markovian jumping complex networks (MJCNs) with non-identical nodes and impulsive effects. Sufficient conditions for the MJCNs are presented based on an M-matrix technique, Lyapunov function method, stochastic analysis technique, and suitable comparison systems to guarantee finite-time synchronization. At last, numerical examples are exploited to illustrate our theoretical results, and they testify the effectiveness of our results for complex dynamic systems.

## 1. Introduction

Over the past decades, owing to the wide applications in engineering and science, for instance, the Internet, power grid networks and communication networks, the search efforts of complex networks have improved considerably [1,2,3,4,5]. Complex networks consist of a great deal of interconnected nodes. As a basic unit, every node has special dynamic behavior different from other nodes. Different types of complex networks have different types of connections. Besides, whether or not the nodes have weight also determines the types of complex networks. The complexity of complex networks is not only manifested in the topological, but also reflected in dynamic movement of the network nodes. Because of the complexity of the network, it has given rise to wide attention from scholars in different fields. In particular, as a hot topic, the synchronization of network nodes has been studied by researchers in kinds of fields [6,7,8,9].

Many publications have presented results concerning asymptotic synchronization of complex networks [9,10,11,12,13,14,15]. Only when the time reaches infinity can the state of the coupling system reach synchronization. That is called asymptotic synchronization. However, asymptotic synchronization is not keeping with practical logic because apparatus and human beings have a limited life span. Consequently, convergence rate is a greatly important index when we study synchronization of complex networks. Therefore, the faster a complex network achieves synchronization, the less resources it consumes.

A notably effective method for achieving faster synchronization is to use a finite-time control technique. The finite-time control technique has several advantages [16,17]. In a settling time, this method can synchronize all nodes in the network. Finite-time control has not only improved robustness of systems but also ameliorated disturbance rejection properties [18,19,20,21,22]. As a result, the finite-time control technique is a more desirable method for achieving synchronization, and it has been widely studied.

In real applications, it is noted that communication signals may be suddenly significantly changed at one time when signals are transmitted. This phenomenon is called impulsive effects [12,20,23,24]. The effects of impulse may promote synchronization or damage synchronization, or yield no effect on synchronization. Recently, based on the concept of average impulsive interval, in a unified framework, the authors in [12,23] studied the synchronization of complex networks with synchronous and desynchronization impulses, respectively. Because the synchronizing impulses are instrumental in synchronization of dynamic systems, many scholars have designed all kinds of impulsive controllers to achieve synchronization of complex networks [12]. In [25], authors proposed an impulsive control method to realize exponential synchronization of coupled complex networks. However, according to [26,27], the desynchronizing impulses which are called impulsive disturbance should not happen too frequently. Because of the instantaneous perturbations, the influence of impulsive disturbance is not negligible. Actually, based on analysis about the stability of impulsive system, a great of results only relate to the asymptotic synchronization, and the finite-time synchronization is less covered. Therefore, studying the finite-time synchronization of dynamic systems with impulsive effects is of vital importance to understand the behaviour of many dynamic networks in real life.

In addition, as a suitable mathematical model for describing a kind of complex network in which the topology is stochastic variation all of a sudden, Markovian jumping complex networks (MJCNs) have given rise to extensive attention among scholars [27,28]. Furthermore, considering that there is a kind of complex dynamic network with finite modes, these modes can switch from one to another at a certain moment. As a result, we consider that it can be looked upon as the frame of theory of the Markovian switching system. In this framework, the Markovian chain controls the switching between different nodes. Many interesting results about synchronization with Markovian chain networks have been reported [29,30,31,32]. In [30,31], the authors indicate that the complex dynamic networks with the Markov jumping topology structure can realize asymptotic synchronization. In [32], authors investigated a stability problem of a class of discrete-time Markovian jumping system via state and mode feedback control. To our best knowledge, by introducing uncertain parameters, the influence of uncertainties in Markovian chains is overcome by a typical method that results in dependence on uncertain-parameter norm of asymptotic synchronization [29,30,31]. However, we need to synchronize the network in finite time in real world. Hence, it is of great importance that we study the finite-time synchronization of MJCNs to meet the needs of the real world.

Note that amounts of published works involved the synchronization of complex dynamic networks usually assume that a series of nodes in the complex dynamic network are identical. From the view of real world, this assumption is impractical for complex networks including all kinds of nodes that usually have different physical parameters [33]. As far as we know, in the previous literature, authors have extensively studied synchronization problems of complex networks with non-identical nodes [34,35,36]. In [34], impulsive complex networks with delayed nonidentical nodes were investigated by Razumikhin technique, a convex combination technique and time varying Lyapunov function. In [36], the authors designed a set of new controllers to achieve finite-time synchronization of dynamic systems with non-identical discontinuous nodes. However, it is remarkable that the problems of finite-time synchronization of non-identity-nodes systems with impulsive effects and markovian jumping topology has not been appropriately investigated.

Motivated by these ideas, it is our task to discuss the finite-time synchronization of Markovian jumping complex dynamic networks with non-identical nodes and impulsive effects. What’s more, a summary of the main contributions in this paper can be listed as follows:The article considers a class of Markovian jumping complex dynamic networks with non-identical nodes and impulsive effects. The system model is more comprehensive and closer to engineering practice. The finite-time method has such outstanding disturbance rejection capability that the results of this subject are of great significance.We propose a new one-norm-based Lyapunov function to solve the difficult points induced by non-identical nodes and impulsive effects. Also, we use the monotonicity to analyze the finite-time synchronization instead of the traditional theorem, and settling time can be theoretically estimated for a given network.Without drawing into any uncertain parameters, the finite time synchronization of dynamic systems is guaranteed by using the stochastic analysis technique, the M-matrix technique and some effective conditions.

The rest of this article is organized as follows. In Section 2, model description and preliminaries are presented. Section 3 introduces the problem of finite-time synchronization of MJCNs with non-identical nodes and impulsive effects, and the convergence analysis is fully derived. In Section 4, we provide a detailed numerical simulations to demonstrate the effectiveness of results. The conclusion is drawn in the last.

Notations: There are some standard notations throughout this paper. $1_{n}$ is a column vector with all the elements being invariant constant 1. $R^{n\times m}$ denotes the set of all n×m matrices and $R^{n}$ is the *n*-dimensional Euclidean space. ${∥\cdot ∥}_{1}$ represents the one-norm of a matrix or a vector, i.e., ${∥x∥}_{1}=\sum_{i=1}^{n}\left|x_{i}\right|$ for $x={\left(x_{1},x_{2},\ldots ,x_{n}\right)}^{T}\in R^{n}$ and, ${∥A∥}_{1}=\max_{j}\sum_{i=1}^{n}\left|a_{ij}\right|$ for $A={\left(a_{ij}\right)}_{n\times n}\in R^{n\times n}$. $\mathrm{sgn}\left(\cdot \right)$ denotes the sign function. $N_{+}$ denotes the set of positive integers and real numbers. Let $\left(\Omega ,F,{\left\{F_{t}\right\}}_{t⩾0},P\right)$ be a complete probability space with filtration ${\left\{F_{t}\right\}}_{t⩾0}$ satisfying the usual conditions (i.e., the filtration contains all P-null sets and is right continuous), and $E\left\{\cdot \right\}$ represents the mathematical expectation operator with respect to a given probability measure P.

## 2. Model Description and Preliminaries

The following preliminaries and necessary assumptions about the system are used throughout this article in this section. Next, we briefly outline the problem formulation. Let $\left\{\sigma (t),t\geq 0\right\}$ be a right continuous Markovian chain on the probability space $\left(\Omega ,F,{\left\{F_{t}\right\}}_{t⩾0},P\right)$ which takes values in the finite state space S=1,2,…,s with generator $\Theta ={\left(\theta_{ij}\right)}_{s\times s}\left(i,j\in S\right)$ given by $$P\left\{\sigma \left(t+\Delta t\right)=j|\sigma \left(t\right)=i\right\}=\left\{\begin{array}{c} \theta_{ij}\Delta t+o\left(\Delta t\right),i\neq j,\\ 1+\theta_{ii}\Delta t+o\left(\Delta t\right),i=j, \end{array}\right.$$
where Δt>0 and $\lim_{\Delta t\to 0}\frac{o\left(\Delta t\right)}{\Delta t}=0$, $\theta_{ij}>0$ is the transfer rate from *i* to *j* if j≠i, yet $\theta_{ii}=-\sum_{j\neq i}\theta_{ij}<0$. The matrix Θ is assumed to be irreducible as a typical hypothesis. Another way to think about this amounts to the situation that, for any i,j∈S, we could get $i_{1},i_{2},\ldots ,i_{k}\in S$ and $j_{1},j_{2},\ldots ,j_{k}\in S$ such that $\theta_{i_{1}j_{1}},{\theta_{i}}_{{}_{2}j_{1}},\ldots ,\theta_{i_{k}j_{1}},{\theta_{i}}_{{}_{k}j_{2}},\ldots ,{\theta_{i}}_{{}_{k}j_{k}}$ are all positive.

Consider the following nonlinear coupled dynamic systems consisting of *N* non-identical nodes with Markovian jumping. The *i*th node is an *n*-dimensional coupled dynamic system, i.e., (1)$${\dot{x}}_{i}\left(t\right)=f_{i}\left(t,x_{i}\left(t\right)\right)+\sum_{j=1}^{N}c_{ij}\left(\sigma \left(t\right)\right)\Phi x_{j}\left(t\right),$$
where $x_{i}\left(t\right)={\left(x_{i1}\left(t\right),x_{i2}\left(t\right),\ldots ,x_{in}\left(t\right)\right)}^{T}\in R^{n}$ is the state of the *i*th node at time *t*, $f_{i}\left(t,x_{i}\left(t\right)\right)={\left(f_{i1}\left(t,x_{i}\left(t\right)\right),f_{i2}\left(t,x_{i}\left(t\right)\right),\ldots ,f_{in}\left(t,x_{i}\left(t\right)\right)\right)}^{T}$ is a continuously nonlinear dynamic function, i=1,2,…,N; and $C\left(\sigma \left(t\right)\right)={\left(c_{ij}\left(\sigma \left(t\right)\right)\right)}_{N\times N}\in R{}^{N\times N}$ is the outer coupling matrix of complex dynamic network, which is defined as follows: if there is a connection between node *j* and node i(i≠j) at time *t*, then $c_{ij}\left(\sigma \left(t\right)\right)>0$, otherwise, $c_{ij}\left(\sigma \left(t\right)\right)=0$. We give a definition about the diagonal elements as $c_{ii}\left(\sigma \left(t\right)\right)=-\sum_{j=1,j\neq i}^{N}c_{ij}\left(\sigma \left(t\right)\right)$. {σ(t),t≥0} is the Markovian process in continuous time, which is represented by the evolution of the node at time *t*. $\Phi =diag\left(\phi_{1},\phi_{2},\ldots ,\phi_{n}\right)\in R^{n\times n}$ is an inner coupling matrix with $\phi_{l}\geq 0,l=1,2,\ldots ,n$.

The main purpose of this article is to synchronize all of the non-identical nodes of system (1) with the following isolate node:(2)$$\dot{z}\left(t\right)=g\left(z\left(t\right)\right)$$
with the initial value z(0), where $z\left(t\right)=(z_{1}\left(t\right),z_{2}\left(t\right),\ldots ,$
$z_{n}{\left(t\right))}^{T}\in R^{n}$ denotes state of the given isolated node, and $g\left(z\left(t\right)\right)={\left(g_{1}\left(z\left(t\right)\right),g_{2}\left(z\left(t\right)\right),\ldots ,g_{n}\left(z\left(t\right)\right)\right)}^{T}$ is a differentiable nonlinear dynamic function with continuity.

Furthermore, in the signal transmission process, signals may be suddenly changed at some discrete time [12,23]. The state $x_{i}\left(t\right),i=1,2,\ldots ,N$ with step change can be represented as a class of differential equation with impulsive effects. Note that the impulsive effects widely found in many processes may potentially have a significant impact on synchronization. Hence, we consider the controlled complex network with impulsive effects as follows:(3)$$\left\{\begin{array}{cc} {\dot{x}}_{i}\left(t\right) & =f_{i}\left(t,x_{i}\left(t\right)\right)+u_{i}\left(\sigma \left(t\right),t\right)+\sum_{j=1}^{N}c_{ij}\left(\sigma \left(t\right)\right)\Phi x_{j}\left(t\right),t\neq t_{k},\\ \Delta x_{i}\left(t_{k}\right) & =\alpha_{i}^{k}e_{i}\left(t_{k}^{-}\right),t=t_{k},k\in N_{+}, \end{array}\right.$$
where i=1,2,…,N, $\Delta x_{i}\left(t_{k}\right)=x_{i}\left(t_{k}\right)-x_{i}\left(t_{k}^{-}\right)$, $x_{i}\left(t_{k}\right)=x_{i}\left(t_{k}^{+}\right)=\lim_{t\to t_{k}^{+}}x_{i}\left(t\right)$, $x_{i}\left(t_{k}^{-}\right)=\lim_{t\to t_{k}^{-}}x_{i}\left(t\right)$, $\varsigma =\{t_{1},t_{2},\ldots \}$ is an impulsive sequence satisfying $0<t_{1}<t_{2}<\cdots <t_{k-1}<t_{k}<\cdots$, and $\lim_{k\to +\infty }t_{k}=+\infty$, $u_{i}\left(\sigma \left(t\right),t\right)$ is the controller to be designed. $e_{i}\left(t\right)=x_{i}\left(t\right)-z\left(t\right)$, constant $|\alpha_{i}^{k}|\geq 0$ is an impulsive gain. Furthermore, in this paper, it is assumed that the pulse phenomena does not rely on the Markovian chain.

**Remark** **1.**
*Signals may be suddenly changed in the form of impulses during signal transmission at some discrete time $t_{k}$, and the impulsive gains can be different from node to node. Additionally, at different impulsive instants, they may be diverse. Thus, $\Delta x_{i}\left(t_{k}\right)=\alpha_{i}^{k}e_{i}\left(t_{k}^{-}\right)$. Impulsive effects are considered as synchronous impulses and desynchronous impulses in this paper, respectively. For analytical simplicity, we should note the impulsive gains as $|\alpha_{i}^{k}|\geq 0,i\in N,k\in N_{+}$.*


Before presenting the main results, some basic assumptions and definitions are presented as follows:
**Assumption** **1.***For any $z{\left(z_{1},z_{2},\ldots ,z_{n}\right)}^{T}\in R^{n}$ and $\bar{z}({\bar{z}}_{1},{\bar{z}}_{2},\ldots ,$${\bar{z}}_{n}{)}^{T}\in R^{n}$, there exists constant $h_{ij}\geq 0$ such that, for i,j=1,2,…,n*(4)$$\left||f_{i}\left(t,z\right)-f_{i}\left(t,\bar{z}\right)\right||\leq \sum_{j=1}^{n}h_{ij}\left|z_{j}-{\bar{z}}_{j}\right|.$$
**Assumption** **2.***There exist positive constants $M_{ij}$ and $L_{j}$ such that, respectively, for i=1,2,…,N,j=1,2,…,n*(5)$$\left\{\begin{array}{c} \left|f_{ij}\left(z\left(t\right)\right)\right|\leq M_{ij},\\ \left|g_{j}\left(z\left(t\right)\right)\right|\leq L_{j}. \end{array}\right.$$

**Definition** **1**
**([37]).**
*The MJCNs (3) is said to be synchronized onto (2) in finite time by putting in a proper designed controller $u_{i}\left(\sigma \left(t\right),t\right)$, if there exists a constant $T_{1}>0$, where $T_{1}$ relies on the initial state $x\left(0\right)={\left(x_{1}^{T}\left(0\right),\ldots ,x_{N}^{T}\left(0\right)\right)}^{T}$, z(0) and the initial value of the Markovian chain σ(0), such that $\lim_{t\to T_{1}}$$E({∥e_{i}\left(t\right)∥}_{1})=0$ and $E({∥e_{i}\left(t\right)∥}_{1})\equiv 0$ for $t\geq T_{1},i=1,2,\ldots ,N$. $T_{1}$ is called as the settling time in this paper.*


**Definition** **2**
**([23]).**
*Average impulsive interval of impulsive sequence $\varsigma =\{t_{1},t_{2},\ldots \}$ is equivalent to $T_{a}$ if there exists a positive integer $N_{0}$ and a positive constant $T_{a}$ such that*
(6)$$\frac{\tilde{t}-t}{T_{a}}-N_{0}\leq N_{\varsigma }\left(t,\tilde{t}\right)\leq N_{0}+\frac{\tilde{t}-t}{T_{a}},$$
*for any $\tilde{t}>t\geq 0$, $N_{\varsigma }\left(t,\tilde{t}\right)$ represents the number of impulsive times of the impulsive sequence ς on the interval $\left(t,\tilde{t}\right)$.*


**Definition** **3**
**([38]).**
*A nonpositive real matrix $A={\left(a_{ij}\right)}_{n\times n}\in R^{n\times n}$ with $a_{ij}\leq 0$ for i≠j is known as Minkowski matrix (M-matrix for short) if all the eigenvalues of A have positive real parts.*


Some significant lemmas are listed in the following part and they are required to obtain the results in this paper.

**Lemma** **1**
**([38]).**
*The following descriptions are equivalent if $A={\left(a_{ij}\right)}_{n\times n}\in R^{n\times n}$ with $a_{ij}\leq 0\left(i\neq j\right)$:*
*(1)* 
*All the eigenvalues of A have positive real parts.*
*(2)* 
*A is a nonsingular Minkowski matrix (M-matrix).*
*(3)* 
*$A^{-1}$ exists and all the elments of $A^{-1}$ are nonnegative.*



**Lemma** **2**
**([37]).**
*Let $A={\left(a_{ij}\right)}_{n\times n}\in R^{n\times n}$ with $a_{ij}\leq 0\left(i\neq j\right)$, $\sum_{j=1}^{n}a_{ij}=0$, $v=diag(v_{1},v_{2},\ldots ,v_{n})$, with $v_{i}>0,i,j=1,2,\ldots ,n$. The A+v is a nonsingular M-matrix if matrix A is irreducible.*


**Lemma** **3**
**([37]).**
*Suppose $\beta_{i\sigma }>\chi_{i\sigma }$, where $\beta_{i\sigma }>0,i=1,2,\ldots ,N,$σ∈S, we obtain $\xi_{\sigma }=\max\left\{\chi_{i\sigma }-\beta_{i\sigma },i=1,2,\ldots ,N\right\}<0$. Denote $\xi =diag(\xi_{1},\xi_{2},\ldots ,\xi_{s})$. The new matrix −Θ−ξ is a nonsingular M-matrix according to Lemma 2 and the irreducibility of *Θ*. All the elements of ${\left(-\Theta -\xi \right)}^{-1}$ are nonnegative because of Lemma 1. At least one positive element exists in each row of ${\left(-\Theta -\xi \right)}^{-1}$ since ${\left(-\Theta -\xi \right)}^{-1}$ is an invertible matrix,. λ is denoted to the maximum of the row sums of ${\left(-\Theta -\xi \right)}^{-1}$. Then, all the elements of ${\left(\rho_{1},\rho_{2},\ldots ,\rho_{s}\right)}^{T}=\frac{1}{\lambda }{\left(-\Theta -\xi \right)}^{-1}1_{s}$ are positive and*
(7)$$\sum_{l\in S}\theta_{\sigma l}\rho_{l}+\rho_{\sigma }\xi_{\sigma }=-\frac{1}{\lambda }<0,\sigma \in S.$$


## 3. Finite-Time Synchronization of the Complex Networks

In this section, a theorem is presented to synchronize the MJCNs (3) with non-identical nodes and impulsive disturbances onto the system (2). Moreover, the setting time is theoretically estimated.

For notation simplicity, we define $c_{ij}\left(\sigma \left(t\right)\right)=c_{ij\sigma }$ and $u_{i}(\sigma \left(t\right),$
$t)=u_{i\sigma }\left(t\right)$, where σ(t)=σ∈S.

Subtracting Equation (2) from Equation (3), we obtain the error dynamical system as follows:(8)$$\left\{\begin{array}{cc} {\dot{e}}_{i}\left(t\right) & =F_{i}\left(t,e_{i}\left(t\right)\right)+\Gamma_{i}\left(t\right)+\sum_{j=1}^{N}c_{ij\sigma }\Phi e_{j}\left(t\right)+u_{i\sigma }\left(t\right),t\neq t_{k},\\ e_{i}\left(t_{k}\right) & =\left(1+\alpha_{i}^{k}\right)e_{i}\left(t_{k}^{-}\right),t=t_{k},k\in N_{+}, \end{array}\right.$$
where $F_{i}\left(t,e_{i}\left(t\right)\right)=f_{i}\left(t,x_{i}\left(t\right)\right)-f_{i}\left(t,z\left(t\right)\right)$, $\Gamma_{i}\left(t\right)=f_{i}\left(t,z\left(t\right)\right)-g\left(t,z\left(t\right)\right)$.

To achieve the aim of finite-time synchronization, we propose the following controller for node *i*:(9)$$u_{i\sigma }\left(t\right)=\left\{\begin{array}{cc} -\beta_{i\sigma }e_{i}\left(t\right)-\eta \frac{e_{i}\left(t\right)}{{∥e(t)∥}_{1}},{∥e(t)∥}_{1} & \neq 0,\\ 0,{∥e(t)∥}_{1} & =0, \end{array}\right.$$
where $\beta_{i\sigma }>0,i=1,2,\ldots ,N,\sigma \in S$ are determined constants, η>0 is an adjustable parameter, and $e\left(t\right)=(e_{1}^{T}\left(t\right),e_{2}^{T}\left(t\right),\ldots ,$
$e_{N}^{T}{\left(t\right))}^{T}$.

**Theorem** **1.**
*For system (3) with controller (9), suppose that Assumptions 1 and 2 hold. The average impulsive interval of the impulsive sequence $\varsigma =\{t_{1},t_{2},\ldots \}$ is $T_{a}$. Then, the following conditions are satisfied:*
(10)$$\beta_{i\sigma }>{∥H_{i}∥}_{1}+c_{ii\sigma }\underline{\phi }+\sum_{j=1,j\neq i}^{N}c_{ji\sigma }\bar{\phi }\equiv \chi_{i\sigma },$$
(11)$$\eta >\sum_{i=1}^{N}\sum_{j=1}^{n}\left(M_{ij}+L_{j}\right),$$
*for σ∈S,i=1,2,…,N, it can be concluded that:*
*(1)* 
*when α>1, and*
(12)$$T_{a}>\left(\frac{\alpha^{2N_{0}}\ln\alpha }{\underline{\rho }\eta^{*}}\right)v\left(0\right),$$
*the complex dynamic networks (3) can achieve synchronization with systems (2) in finite time, and the settling time is*
(13)$$T_{1}=\frac{T_{a}}{\ln\alpha }\ln\left(\frac{\underline{\rho }\eta^{*}T_{a}}{\underline{\rho }\eta^{*}T_{a}-\alpha^{2N_{0}}v\left(0\right)\ln\alpha }\right);$$
*(2)* 
*when α=1, the complex dynamic networks (3) is synchronized onto systems (2) in finite time, and the settling time is*
(14)$$T_{1}=\frac{v\left(0\right)}{\rho \eta^{*}};$$
*(3)* 
*when 0<α<1, the complex dynamic networks (3) is synchronized onto systems (2) within finite time, and the settling time is*
(15)$$T_{1}=\frac{T_{a}}{\ln\alpha }\ln\left(\frac{\underline{\rho }\eta^{*}T_{a}}{\underline{\rho }\eta^{*}T_{a}-\alpha^{-2N_{0}}v\left(0\right)\ln\alpha }\right);$$
*where $H_{i}={\left(h_{jk}^{i}\right)}_{n\times n}$, $h_{jk}^{i}$ represents the Lipschitz coefficient of ith node, $\underline{\phi }=\min\left\{\phi_{b},b=1,2,\ldots ,n\right\},\bar{\phi }=\max\{\phi_{b},$b=1,2,…,n}, $\eta^{*}=\eta -\sum_{i=1}^{N}\sum_{j=1}^{n}\left(M_{ij}+L_{j}\right)$, $\underline{\rho }=\min\left\{\rho_{\sigma },\sigma \in S\right\}$, defining $0<\rho_{\sigma }<1$ has been referred to cursorily in Lemma 3, $v\left(0\right)=\rho_{\sigma (0)}\sum_{i=1}^{N}{∥e_{i}\left(0\right)∥}_{1}$. σ(0) is the initial value of the Markovian chain, $\alpha \geq |1+\alpha_{i}^{k}|,i=1,2,\ldots ,N$.*



**Proof.** Construct the following Lyapunov function candidate with Markovian switch for σ(t)=σ∈S: (16)$$V\left(e\left(t\right),\sigma ,t\right)=\rho_{\sigma }\sum_{i=1}^{N}{∥e_{i}\left(t\right)∥}_{1}.$$We use LV(e(t),σ(t),t) to represent the infinitesimal operator of V(e(t),σ(t),t). When ${∥e(t)∥}_{1}\neq 0$, according to It$\hat{o}$ differential formula, it is differentiated from V(e(t),σ(t),t) along the trajectory of (8), which gives that, for $t\neq_{k},k\in N_{+}$
(17)$$\begin{array}{cc} LV(e(t),\sigma ,t)= & \rho_{\sigma }\sum_{i=1}^{N}1_{n}^{T}diag\left(\mathrm{sgn}\left(e_{i}\left(t\right)\right)\right)\times [F_{i}\left(e_{i}\left(t\right)\right)+\Gamma_{i}\left(t\right)+\sum_{j=1}^{N}c_{ij\sigma }\Phi e_{j}\left(t\right)\\ & -\beta_{i\sigma }e_{i}\left(t\right)-\eta \frac{e_{i}\left(t\right)}{{∥e(t)∥}_{1}}]+\sum_{l\in S}^{N}\theta_{\sigma l}\rho_{l}\sum_{i=1}^{N}{∥e_{i}\left(t\right)∥}_{1}, \end{array}$$
where $sgn\left(e_{i}\left(t\right)\right)={\left(sgn\left(e_{i1}\left(t\right)\right),sgn\left(e_{i2}\left(t\right)\right),\ldots ,sgn\left(e_{in}\left(t\right)\right)\right)}^{T}$.Based on Assumption 1, one obtains (18)$$1_{n}^{T}diag\left(\mathrm{sgn}\left(e_{i}\left(t\right)\right)\right)F_{i}\left(e_{i}\left(t\right)\right)\leq {∥H_{i}∥}_{1}{∥e_{i}\left(t\right)∥}_{1}.$$Note that in Assumption 2, (19)$$1_{n}^{T}diag\left(\mathrm{sgn}\left(e_{i}\left(t\right)\right)\right)\Gamma_{i}\left(t\right)\leq \sum_{j=1}^{n}M_{ij}+\sum_{j=1}^{n}L_{j},$$
and substituting (18) and (19) into (17) yields (20)$$\begin{array}{cc} LV(e(t),\sigma ,t)\leq & \sum_{j=1}^{N}[\rho_{\sigma }{∥H_{i}∥}_{1}{∥e_{i}\left(t\right)∥}_{1}+\rho_{\sigma }\sum_{j=1}^{n}(M_{ij}+L_{j})+\sum_{j=1}^{N}\rho_{\sigma }c_{ij\sigma }\Phi e_{j}\left(t\right)\\ & +\rho_{\sigma }\left(-\beta_{i\sigma }e_{i}\left(t\right)+\eta \frac{e_{i}\left(t\right)}{{∥e(t)∥}_{1}}\right)]+\sum_{l\in S}^{N}\theta_{\sigma l}\rho_{l}\sum_{i=1}^{N}{∥e_{i}\left(t\right)∥}_{1}. \end{array}$$After careful planning and analysis, we find that (21)$$\begin{array}{c} LV\left(e\left(t\right),\sigma ,t\right)\leq \sum_{i=1}^{N}[(\sum_{l\in S}^{N}\theta_{\sigma l}\rho_{l}+\rho_{\sigma }\left(\chi_{i\sigma }-\beta_{i\sigma }\right)){∥e_{i}\left(t\right)∥}_{1}-\left(\rho_{\sigma }\eta -\rho_{\sigma }\sum_{j=1}^{N}\left(M_{ij}+L_{j}\right)\right)]. \end{array}$$By Lemma 3, it follows from inequality (7) that $\sum_{l\in S}\theta_{\sigma l}\rho_{l}$
$+\rho_{\sigma }+\rho_{\sigma }\xi_{\sigma }=-\frac{1}{\lambda }<0,\sigma \in S,i=1,2,\ldots ,N$. Therefore, according to (7) and (21), we can obtain that (22)$$LV\left(e\left(t\right),\sigma ,t\right)\leq -\underline{\rho }\left[\eta -\sum_{i=1}^{N}\sum_{j=1}^{n}\left(M_{ij}+L_{j}\right)\right].$$Let $\eta^{*}=\eta -\sum_{i=1}^{N}\sum_{j=1}^{n}\left(M_{ij}+L_{j}\right)$,therefore, it is obtained that (23)$$LV\left(e\left(t\right),\sigma ,t\right)\leq -\underline{\rho }\eta^{*}<0.$$Based on the arbitrariness of σ∈S, by taking mathematical expectation, it follows from (23) that (24)$$\frac{d}{dt}E\left\{V\left(e\left(t\right),\sigma \left(t\right),t\right)\right\}\leq -\underline{\rho }\eta^{*}.$$When $t=t_{k}$, we can get from (16) and the second equation of (8) that (25)$$V\left(e\left(t_{k}^{+}\right),\sigma \left(t_{k}^{+}\right),t_{k}^{+}\right)\leq \alpha V\left(e\left(t_{k}^{-}\right),\sigma \left(t_{k}^{-}\right),t_{k}^{-}\right).$$Consider the following comparison system: (26)$$\left\{\begin{array}{c} \dot{v}\left(t\right)=\left\{\begin{array}{c} -\underline{\rho }\eta^{*},v\left(t\right)>0,t\neq t_{k}\\ 0,v\left(t\right)=0,t\neq t_{k} \end{array}\right.\\ v\left(t_{k}^{+}\right)=\alpha v\left(t_{k}^{-}\right),t=t_{k},k\in N_{+}\\ v\left(0\right)=\rho_{\sigma (0)}\sum_{i=1}^{N}{∥e_{i}\left(0\right)∥}_{1}. \end{array}\right.$$Consider e(t)=0 is an equilibrium point of the error system (8),when ${∥e(t)∥}_{1}=0$, we find that $\frac{d}{dt}E\left\{V\left(e\left(t\right),\sigma \left(t\right),t\right)\right\}=0$ which is equivalent to $\dot{v}\left(t\right)=0$ in (26). Therefore, comparing (24) and (25) with (26), one can see that V(e(0),σ(0),
0)=v(0) indicates 0≤E{V(e(t),σ(t),t)}≤v(t) for t≥0. According to the Squeeze Theorem, we can derive that E{V(e(t),
σ(t),t)}=0 if v(t)=0.By using reduction to absurdity, we can prove Theorem 1 properly. If there exists $T_{1}\in \left[0,+\infty \right)$ such that $v(T_{1})=0$, then it is obtained from (26) that v(t)≡0 for all $t\geq T_{1}$. If not, there exists $T_{2}>T_{1}$ such that $v(T_{2})>0$. Let $T_{r}=\sup\left\{t\in \left[T_{1},T_{2}\right]:v\left(t\right)=0\right\}$, then $T_{r}<T_{2},v\left(T_{r}\right)=0$, and v(t)>0 for all $t\in (T_{r},T_{2}]$. Moreover, v(t) is monotonic increasing on the interval $\left[T_{r},T_{3}\right]$ because of existing $T_{3}\in \left(T_{r},T_{2}\right]$, i.e., $\dot{v}\left(t\right)>0$ for $t\in [T_{r},T_{2}]$, which contradicts the first equation of (26). In the meantime, $E\{V\left(e\left(T_{1}\right),\sigma \left(T_{1}\right),T_{1}\right)\}=0$ and E{V(e(t),σ(t),t)}≡0 for all $t\geq T_{1}$, which indicates that $\lim_{t\to T_{1}}E\left({∥e_{i}\left(t\right)∥}_{1}\right)=0$ and $E({∥e_{i}\left(t\right)∥}_{1})\equiv 0$ for $t\geq T_{1},i=1,2,\ldots ,N$.The value $\alpha_{i}^{k}$ will lead to α>1,α=1,0<α<1. Next, we give the provement about estimating the settling time $T_{1}$, and there are three cases to be considered in the following part.When v(t)>0, according to (26), we can deduce that when $t\in \left[t_{0}^{+},t_{1}^{-}\right)$, it yields that $v\left(t\right)=v\left(t_{0}^{+}\right)-\rho \eta^{*}\int_{t_{0}^{+}}^{t}dt$.Similarly, for $t\in \left[t_{k}^{+},t_{k+1}^{-}\right)$, one obtains that $v\left(t\right)=\alpha^{k}v\left(0\right)-\rho \eta^{*}\int_{t_{0}}^{t_{1}}\alpha^{k}dt-\rho \eta^{*}\int_{t_{1}}^{t_{2}}\alpha^{k-1}dt-\cdots -\rho \eta^{*}\int_{t_{k}}^{t}dt$.By iteration, the above equations are incorporated as follows: (27)$$v\left(t\right)=\alpha^{N_{\varsigma }\left(0,t\right)}v\left(0\right)-\underline{\rho }\eta^{*}\int_{0}^{t}\alpha^{N_{\varsigma }\left(y,t\right)}dy.$$In the case of α>1 it can be deduced on the basis of (6) and (27) that (28)$$\begin{array}{cc} v(t)\leq & \left(\alpha^{N_{0}}v\left(0\right)-\underline{\rho }\eta^{*}\alpha^{-N_{0}}\frac{T_{a}}{\ln\alpha }\right)e^{\frac{\ln\alpha }{T_{a}}t}+\underline{\rho }\eta^{*}\alpha^{-N_{0}}\frac{T_{a}}{\ln\alpha }. \end{array}$$Considering (12) and the fact that lnα>0, the right-hand side of (28) becomes zero because of existing $T_{1}\in \left(0,\infty \right)$ and (29)$$T_{1}=\frac{T_{a}}{\ln\alpha }\ln\left(\frac{\underline{\rho }\eta^{*}T_{a}}{\underline{\rho }\eta^{*}T_{a}-\alpha^{2N_{0}}v\left(0\right)\ln\alpha }\right).$$When α=1, it can be obtained from (27) that $T_{1}=v\left(0\right)/\rho \eta^{*}$.Similarly, when 0<α<1, we can get that (30)$$\begin{array}{cc} v(t)\leq & \left(\alpha^{-N_{0}}v\left(0\right)-\underline{\rho }\eta^{*}\alpha^{N_{0}}\frac{T_{a}}{\ln\alpha }\right)e^{\frac{\ln\alpha }{T_{a}}t}+\underline{\rho }\eta^{*}\alpha^{N_{0}}\frac{T_{a}}{\ln\alpha }. \end{array}$$Since $\alpha^{-N_{0}}v\left(0\right)-\rho \eta^{*}\alpha^{N_{0}}\left(T_{a}/\ln\alpha \right)>0$, $\left(\ln\alpha /T_{a}\right)<0$, and $\rho \eta^{*}\alpha^{N_{0}}\left(T_{a}/\ln\alpha \right)<0$, the right-hand side of (30) becomes zero due to exist $T_{1}\in \left(0,+\infty \right)$, and (31)$$T_{1}=\frac{T_{a}}{\ln\alpha }\ln\left(\frac{\underline{\rho }\eta^{*}T_{a}}{\underline{\rho }\eta^{*}T_{a}-\alpha^{-2N_{0}}v\left(0\right)\ln\alpha }\right).$$Based on the above analysis, we can obtain that $\lim_{t\to T_{1}}$
$E({∥e_{i}\left(t\right)∥}_{1})=0$ and $E({∥e_{i}\left(t\right)∥}_{1})\equiv 0$ for $t\geq T_{1},i=1,2,\ldots ,N$. On the basis of Definition 1, the complex dynamic network (2) can realize synchronization with (3) in finite time via the control law (9). This completes the proof of the Theorem 1. □

**Remark** **2.**
*As a classic example, the upper bound and lower bound of impulsive intervals are demonstrated by the Example 3 in the reference [23]. The impulsive sequence $\bar{\zeta }=\left\{t_{1},t_{2},\ldots \right\}$ can be expressed as the following equation*
(32)$$t_{k}-t_{k-1}=\left\{\begin{array}{c} \varepsilon ,\;\mathrm{mod}\;(k,N_{0})\neq 0\\ N_{0}\left(T_{a}-\varepsilon \right)+\varepsilon ,\;\mathrm{mod}\;\left(k,N_{0}\right)=0. \end{array}\right.$$
*where ε and $T_{a}$ denotes positive numbers satisfying $\varepsilon \leq T_{a}$; $\;\mathrm{mod}\;(k,N_{0})$ represents the remainder of k dividing by $N_{0}$ and $N_{0}\in N{}_{+}$. According to (32), we can obtain that $\inf_{k\in N{}_{+}}\left\{t_{k}-t_{k-1}\right\}\in \varepsilon$ and $\sup_{k\in N{}_{+}}\left\{t_{k}-t_{k-1}\right\}=N_{0}\left(T_{a}-\varepsilon \right)+\varepsilon$. The quantity $\inf_{k\in N{}_{+}}\left\{t_{k}-t_{k-1}\right\}$ will be quite small, and the quantity $\sup_{k\in N{}_{+}}\left\{t_{k}-t_{k-1}\right\}$ will be very big if ε is small enough and $N_{0}$ is sufficiently large. In such a situation, two types of impulsive effects, namely, synchronizing impulses and desynchronizing impulses are considered in this article. In order to ensure the finite-time synchronization desynchronizing impulses shouldn’t occur frequently. Given the analysis provided above, our criterion in Theorem 1 is proper for impulsive control signals which have a wider range. Compared with the results deriving by $\sup_{k\in N{}_{+}}\left\{t_{k}-t_{k-1}\right\}$, our results are less conservative. To guarantee that the impulses do not occur too frequently, the positive integer $N_{0}$ will not be too large. Accordingly, the impulsive interval $T_{a}$ will not be too small.*


It is worth noting that our work is affected by the maximum average impulsive intervals and the minimum average impulsive intervals. Are the networks (3) synchronized to the isolate system (2) in finite-time with the maximum average impulsive intervals and the minimum average impulsive intervals? To answer these questions it is worthwhile to investigate the influence of the maximum average and the minimum average impulsive intervals.

Respectively, we replace $T_{a}$ with $\underline{T}$ and $\bar{T}$, and the inequality (6) will be adapted to the following form (33)$$\frac{t-s}{\bar{T}}-1\leq N_{\varsigma }\left(s,t\right)\leq \frac{t-s}{\underline{T}}+1.$$
for any t>s≥0, where $\underline{T}=inf_{k\in N_{+}}\left\{t_{k+1}-t_{k}\right\},\bar{T}=sup_{k\in N_{+}}$
$\left\{t_{k+1}-t_{k}\right\}$.

**Corollary** **1.**
*For system (3) with controller (9), assume that Assumptions 1 and 2 hold. The minimum impulsive interval and the maximum impulsive interval of impulsive signals $\varsigma =\{t_{1},t_{2},\ldots \}$ are $\underline{T}$ and $\bar{T}$, respectively. There exists a positive number α such that $1<|1+\alpha_{i}^{k}|\leq \alpha ,i\in 1,2,\ldots ,N$, and satisfied conditions are listed as follows:*
(34)$$\beta_{i\sigma }>{∥H_{i}∥}_{1}+c_{ii\sigma }\underline{\phi }+\sum_{j=1,j\neq i}^{N}c_{ji\sigma }\bar{\phi }\equiv \chi_{i\sigma },$$
(35)$$\eta >\sum_{i=1}^{N}\sum_{j=1}^{n}\left(M_{ij}+L_{j}\right),$$
(36)$$\underline{T}>\left(\frac{\alpha^{2}\ln\alpha }{\underline{\rho }\eta^{*}}\right)v\left(0\right),$$
*for σ∈S,i=1,2,…,N, then complex dynamical network (3) with controller (9) is synchronized onto system (2) in a finite time $T_{1}$, where $H_{i}={\left(h_{jk}^{i}\right)}_{n\times n},\underline{\phi }=\min\left\{\phi_{b},b=1,2,\ldots ,n\right\},\bar{\phi }=\max\{\phi_{b},$b=1,2,…,n}, $\eta^{*}=\eta -\sum_{i=1}^{N}\sum_{j=1}^{n}\left(M_{ij}+L_{j}\right),\underline{\rho }=\min\left\{\rho_{\sigma },\sigma \in S\right\},0<\rho_{\sigma }<1$ is defined in the proof, $v\left(0\right)=\rho_{\sigma (0)}\sum_{i=1}^{N}{∥e_{i}\left(0\right)∥}_{1}$. σ(0) is the initial value of the Markovian chain.*


When v(t)>0, according to (26), we can deduce that (37)$$v\left(t\right)=\alpha^{N_{\varsigma }\left(0,t\right)}v\left(0\right)-\underline{\rho }\eta^{*}\int_{0}^{t}\alpha^{N_{\varsigma }\left(y,t\right)}dy.$$

In the case of α>1, it can be deduced on the basis of (33) and (37) that (38)$$v\left(t\right)\leq \alpha^{\frac{t}{\underline{T}}+1}v\left(0\right)+\underline{\rho }\eta^{*}\alpha^{-1}\frac{\bar{T}}{\ln\alpha }-\underline{\rho }\eta^{*}\alpha^{-1}\bar{T}\frac{\alpha^{\frac{t}{\bar{T}}}}{\ln\alpha }=f\left(t\right).$$

Compute the derivative of f(t)
(39)$$\dot{f}\left(t\right)=\frac{\alpha v\left(0\right)\alpha^{\frac{t}{\underline{T}}}\ln\alpha }{\underline{T}}-\underline{\rho }\eta^{*}\alpha^{-1}\alpha^{\frac{t}{\bar{T}}}.$$

Then, let $\dot{f}\left(\tilde{t}\right)=0$, we can find that (40)$$\tilde{t}=\frac{\bar{T}\underline{T}\ln\frac{\alpha^{2}v\left(0\right)\ln\alpha }{\underline{\rho }\eta^{*}\underline{T}}}{(\underline{T}-\bar{T})\ln\alpha }.$$

Then we can obtain that $$f(\tilde{t})<0.$$

Furthermore, since v(0)>0, we can get f(0)=αv(0)>0.

According to the existence theorem of zero point, for $t\in \left[0,\tilde{t}\right]$, we can get that f(t) has one zero point $T_{1}$.

**Corollary** **2.**
*For system (3) with controller (9), assume that Assumptions 1 and 2 hold. The minimum impulsive interval and the maximum impulsive interval of impulsive signals $\varsigma =\{t_{1},t_{2},\ldots \}$ are $\underline{T}$ and $\bar{T}$, respectively. There exists a positive number exists α such that $|1+\alpha_{i}^{k}|\leq \alpha <1,i\in 1,2,\ldots ,N$, and the satisfied conditions are listed as follows:*
(41)$$\beta_{i\sigma }>{∥H_{i}∥}_{1}+c_{ii\sigma }\underline{\phi }+\sum_{j=1,j\neq i}^{N}c_{ji\sigma }\bar{\phi }\equiv \chi_{i\sigma },$$
(42)$$\eta >\sum_{i=1}^{N}\sum_{j=1}^{n}\left(M_{ij}+L_{j}\right),$$
*for σ∈S,i=1,2,…,N, then complex dynamical network (3) with controller (9) is synchronized onto system (2) in a finite time $T_{1}$, where $H_{i}={\left(h_{jk}^{i}\right)}_{n\times n},\underline{\phi }=\min\left\{\phi_{b},b=1,2,\ldots ,n\right\},\bar{\phi }=\max\{\phi_{b},$b=1,2,…,n}, $\eta^{*}=\eta -\sum_{i=1}^{N}\sum_{j=1}^{n}\left(M_{ij}+L_{j}\right),\underline{\rho }=\min\left\{\rho_{\sigma },\sigma \in S\right\},0<\rho_{\sigma }<1$ is defined in the proof, $v\left(0\right)=\rho_{\sigma (0)}\sum_{i=1}^{N}{∥e_{i}\left(0\right)∥}_{1}$. σ(0) is the initial value of the Markovian chain.*


The proof is similar to that of Corollary 1, so we omit it.

**Remark** **3.**
*Corollary 2 and Theorem 1 indicate that average impulsive interval does not matter to the finite-time synchronization when 0<α<1.*


## 4. Numerical Examples

In this section, numerical simulations are provided to demonstrate that the applications of the theorem described above are valid. Using the similar example as that given in [33], we also consider two different chaotic node systems as follows:(43)$$\begin{array}{cc} \dot{x}\left(t\right) & =\left(\begin{array}{ccc} -0.8 & 6 & 0\\ 0.2 & -0.2 & 0.2\\ 0 & -2.86 & 0 \end{array}\right)x\left(t\right)+\left(\begin{array}{c} 0.9\left|x_{1}\left(t\right)+1\right|-0.9\left|x_{1}\left(t\right)-1\right|\\ 0\\ 0 \end{array}\right)=\bar{f}\left(x\left(t\right)\right), \end{array}$$
(44)$$\begin{array}{cc} \dot{x}\left(t\right) & =\left(\begin{array}{ccc} -0.6 & 1.5 & 0\\ 0.2 & -0.2 & 0.2\\ 0 & -2.6 & 0 \end{array}\right)x\left(t\right)+\left(\begin{array}{c} 0.7\left|x_{1}\left(t\right)+1\right|-0.7\left|x_{1}\left(t\right)-1\right|\\ 0\\ 0 \end{array}\right)=\tilde{f}\left(x\left(t\right)\right), \end{array}$$
where $x\left(t\right)={\left(x_{1}\left(t\right),x_{2}\left(t\right),x_{3}\left(t\right)\right)}^{T}$. Figure 1 and Figure 2 depict the chaotic trajectories of (43) with different initial values $x\left(0\right)={\left(0.5,0.4,0.8\right)}^{T}$ and $x\left(0\right)={\left(0.2,0.4,0.7\right)}^{T}$, respectively. Then, Figure 3 depicts the chaotic trajectories of (44) with the initial value $x\left(0\right)={\left(0.5,0.4,0.8\right)}^{T}$.

In the following, consider a coupled complex network being composed of two types of non-identical chaotic nodes (43) and (44), which are described such that:(45)$$\left\{\begin{array}{cc} {\dot{x}}_{i}\left(t\right) & =f_{i}\left(t,x_{i}\left(t\right)\right)+u_{i}\left(\sigma \left(t\right),t\right)+\sum_{j=1}^{3}c_{ij}\left(\sigma \left(t\right)\right)\Phi x_{j}\left(t\right),t\neq t_{k}\\ \Delta x_{i}\left(t_{k}\right) & =\alpha_{i}^{k}e_{i}\left(t_{k}\right),t=t_{k},k\in N_{+},i=1,2,3. \end{array}\right.$$
where ${\dot{x}}_{i}\left(t\right)={\left(x_{i1}\left(t\right),x_{i2}\left(t\right),x_{i3}\left(t\right)\right)}^{T},f_{i}\left(x_{i}\left(t\right)\right)=\bar{f}\left(x_{i}\left(t\right)\right),$
i=1,2, and $f_{i}\left(x_{i}\left(t\right)\right)=\tilde{f}\left(x_{i}\left(t\right)\right),i=3.\Phi =\mathrm{diag}\left(1,1,0\right),$ and the outer coupling matrix is supposed to be


$C_{1}=0.01\left(\begin{array}{ccc} -2 & 1 & 1\\ 0 & -1 & 1\\ 1 & 2 & -3 \end{array}\right),$
$C_{2}=0.01\left(\begin{array}{ccc} -1 & 0 & 1\\ 1 & -2 & 1\\ 0 & 1.5 & -1.5 \end{array}\right),$
$C_{3}=0.01\left(\begin{array}{ccc} -1.5 & 0.5 & 1\\ 0.5 & -3 & 2.5\\ 1 & 1 & -2 \end{array}\right).$


Moreover, the transition probability matrix is given as $\Theta =\left(\begin{array}{ccc} -40 & 18 & 32\\ 30 & -80 & 50\\ 40 & 30 & -70 \end{array}\right).$

According to Assumption 1, we can verify the matrix which is similar to [33] satisfying with $$H_{1}=H_{2}=\left(\begin{array}{ccc} 4.9 & 1.6 & 0\\ 0.2 & 0.2 & 0.2\\ 0 & 2.9 & 0 \end{array}\right),H_{3}=\left(\begin{array}{ccc} 2.3 & 1.8 & 0\\ 0.1 & 0.1 & 0.1\\ 0 & 12.8 & 0 \end{array}\right).$$

We synchronize (45) onto the following isolated node system:(46)$$\begin{array}{cc} \dot{z}\left(t\right) & =0.2\left(\begin{array}{ccc} -3 & 7 & 0\\ 1 & -1 & 1\\ 0 & -11 & 0 \end{array}\right)z\left(t\right)+0.2\left(\begin{array}{c} 4.5\left|z_{1}\left(t\right)+1\right|-4.5\left|z_{1}\left(t\right)-1\right|\\ 0\\ 0 \end{array}\right)=g\left(z\left(t\right)\right), \end{array}$$
where $z\left(t\right)={\left(z_{1}\left(t\right),z_{2}\left(t\right),z_{3}\left(t\right)\right)}^{T}$. The initial condition is taken as $z\left(0\right)={\left(-0.3,0.1,0.12\right)}^{T}$. Figure 4 shows the chaotic trajectory of (46).

**Remark** **4.**
*In numerical simulations, choosing different initial values under identical chaotic systems and choosing the same initial values under non-identical chaotic systems for comparison can more effectively illustrate the results of this study.*


The same as [33] calculated simply, it can be easily calculated the corresponding parameters as $M_{11}=M_{21}=1.7411,M_{12}=M_{22}=0.6782,$
$M_{13}=M_{23}=3.7841,M_{31}=1.8079,M_{32}=0.7097,M_{33}=3.2991,L_{1}=1.7662,L_{2}=0.6572,L_{3}=2.8854,\chi_{11}=4.866,$
$\chi_{21}=4.886,\chi_{31}=4.38,\chi_{12}=4.866,\chi_{22}=4.871,\chi_{32}=4.38,\chi_{13}=4.871,\chi_{23}=4.871,\chi_{33}=4.395.$ Thus, choosing $\beta_{i\sigma }=4.9,i=1,2,3,\sigma =1,2,3,$ we obtain ξ=−diag(0.034,
0.029,0.029). Furthermore, from ${\left(\rho_{1},\rho_{2},\ldots ,\rho_{s}\right)}^{T}=\frac{1}{\lambda }(-\Theta$
$-\xi^{-1}1_{s}$ we find $\rho_{1}=0.9998,\rho_{2}=1,\rho_{3}=0.9999.$ The initial values are arbitrarily and evenly chosen from (−3,3) and we find $\sum_{i=1}^{3}{∥e_{i}\left(0\right)∥}_{1}=0.1733$.

In the following section, we verify Theorem 1. The parameters in the simulations are taken as step-length at 0.0001. We choose $\alpha_{1}^{k}=\alpha_{2}^{k}=0.1,\alpha_{3}^{k}=0.2$ for any $k\in N_{+}$ because of the impulsive disturbance. Accordingly, we use α=1.2. Additionally, we choose η=43.8981 and $\sigma \left(0\right)=1$. By simple computation, we obtain $\eta^{*}=10$. The condition (12) in Theorem 1 implies that $T_{a}>0.004898$ should be satisfied. Consider $T_{a}=0.0049,N_{0}=5$ According to the Equation (13), the complex network is synchronized with the isolates node within $T_{1}=0.5245$. The synchronization trajectories ${∥e_{i}\left(t\right)∥}_{1},i=1,2,3$ are shown in Figure 5, Figure 6 and Figure 7. The figures depict that the complex network (3) is synchronized with (2) before t=0.09.

**Remark** **5.**
*It is well known that time is not negative, therefore, $T_{a}$ in (12) of Theorem 1 should not be too small. Otherwise, this kind of statements will reflect a real world situation and will not be in synchronization.*


**Remark** **6.**
*In Corollary 1, we choose the same data for the simulation. We obtain the result $\underline{T}>0.00114$, and $T_{a}=0.0049$. From the simulation data, we can get $\inf_{k\in N{}_{+}}\left\{t_{k}-t_{k-1}\right\}\leq 0.0009$. Obviously, our criterion is suitable for a wider range of impulsive signals. Because the impulsive effect discussed in this paper is considered to be disturbance, the impulses should not occur frequently. Comparing the range of $T_{a}$ and $\underline{T}$, we see that an average impulsive interval of $T_{a}$ is better than the average impulsive interval of $\underline{T}$.*


## 5. Conclusions

In this paper, we have studied the finite-time synchronization of the MJCNs with non-identical nodes and impulsive effects. Sufficient conditions for the MJCNs are presented based on the M-matrix technique, the Lyapunov function method, the stochastic analysis technique and suitable comparison systems to ensure the finite-time synchronization of the communication systems. We show that the MJCNs with non-identical nodes and impulsive effects can reach synchronization within a finite time through rigorous mathematical derivation and simulation examples.

In the real world, we realize that a more general class of complex networks does not have complete information. Therefore, we would further investigate how to synchronize the complex networks with incomplete information in finite time.

## Figures and Tables

**Figure 1 entropy-21-00779-f001:**
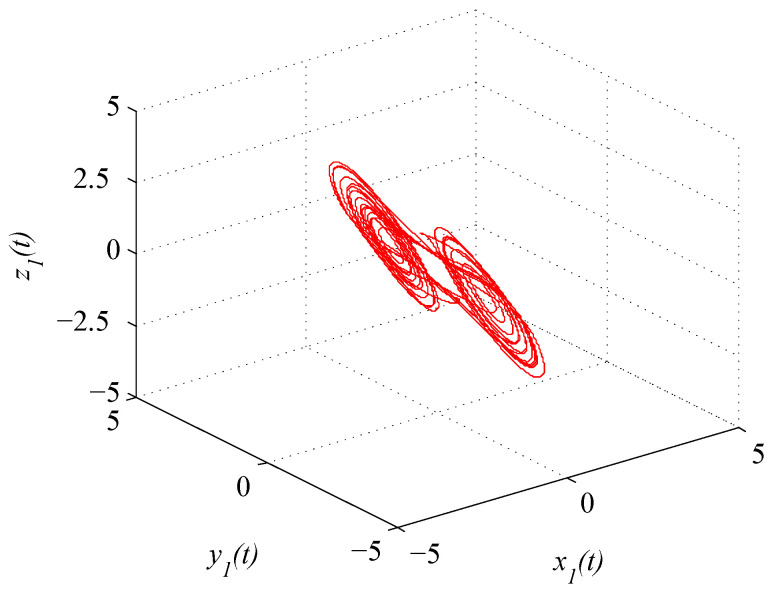
Chaotic trajectory of (43) with initial value $x\left(0\right)={\left(0.5,0.4,0.8\right)}^{T}$.

**Figure 2 entropy-21-00779-f002:**
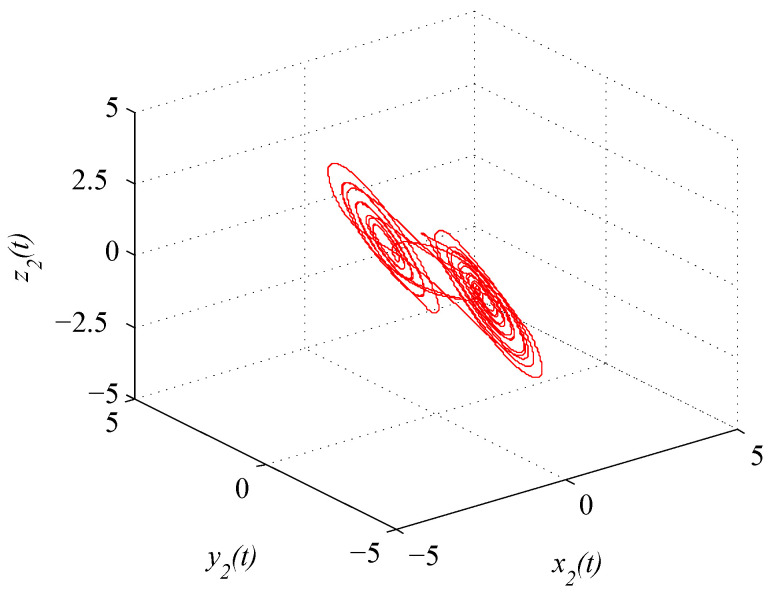
Chaotic trajectory of (43) with initial value $x\left(0\right)={\left(0.2,0.4,0.7\right)}^{T}$.

**Figure 3 entropy-21-00779-f003:**
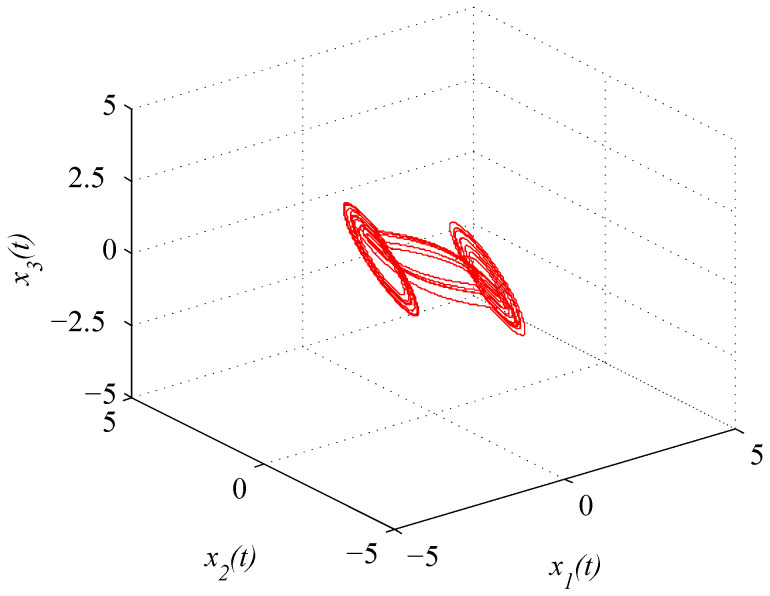
Chaotic trajectory of (44) with initial value $x\left(0\right)={\left(0.5,0.4,0.8\right)}^{T}$.

**Figure 4 entropy-21-00779-f004:**
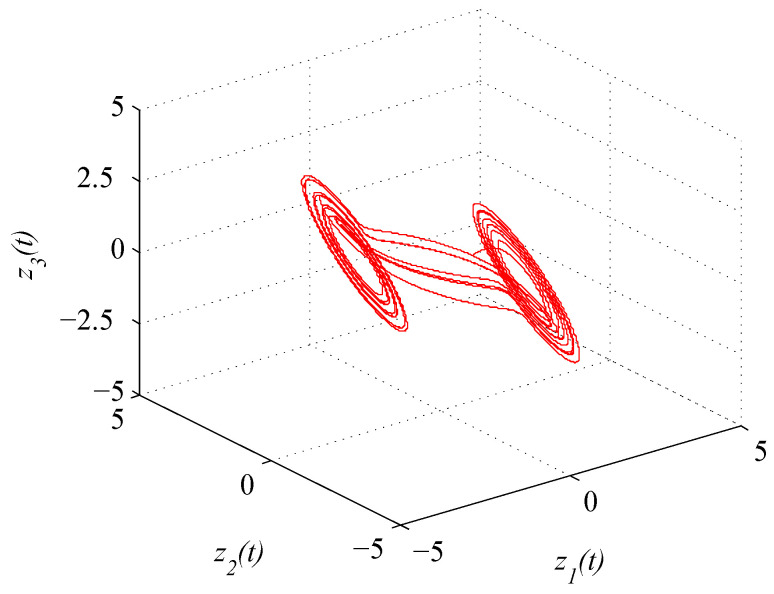
Chaotic trajectory of (46) with initial value $z\left(0\right)={\left(0.3,0.1,0.12\right)}^{T}$.

**Figure 5 entropy-21-00779-f005:**
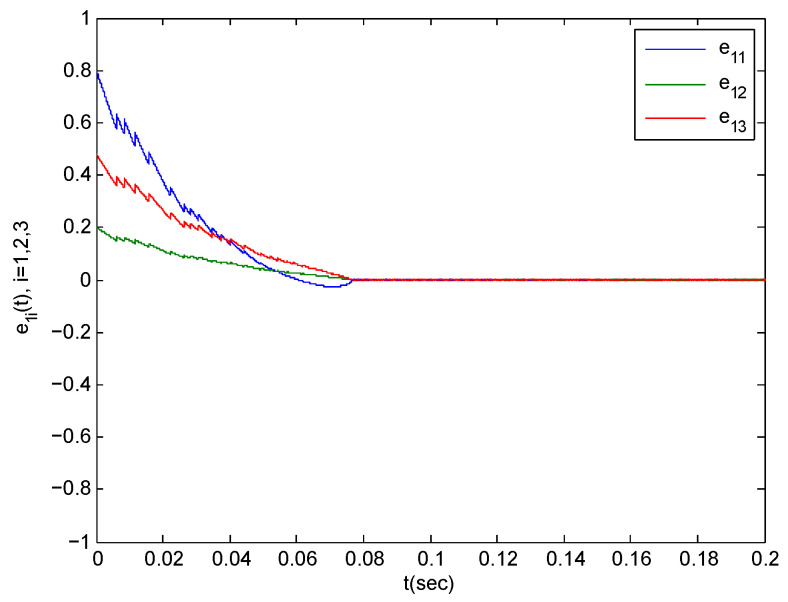
The synchronization error trajectories of complex system (44) with isolated system (46) with an initial value of $x\left(0\right)={\left(0.5,0.4,0.8\right)}^{T}$ under control.

**Figure 6 entropy-21-00779-f006:**
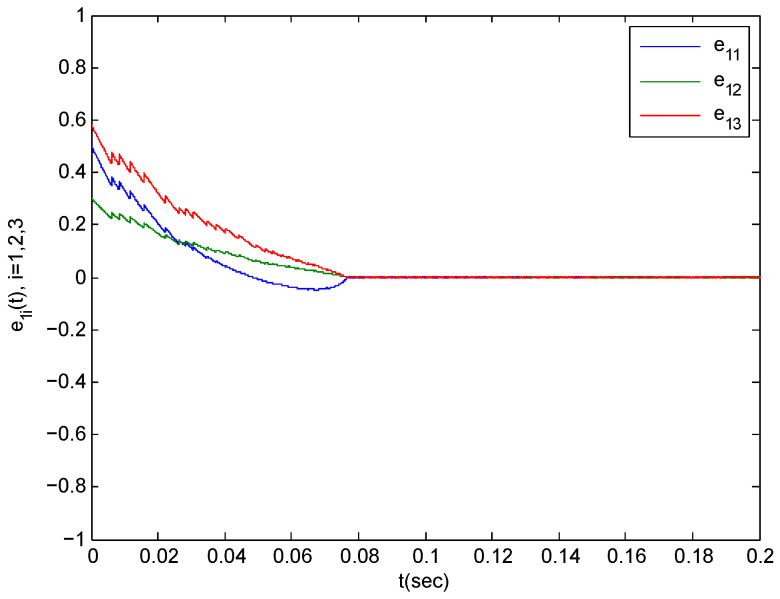
The synchronization error trajectories of complex system (44) with isolated system (46) with an initial value of $x\left(0\right)={\left(0.2,0.4,0.7\right)}^{T}$ under control.

**Figure 7 entropy-21-00779-f007:**
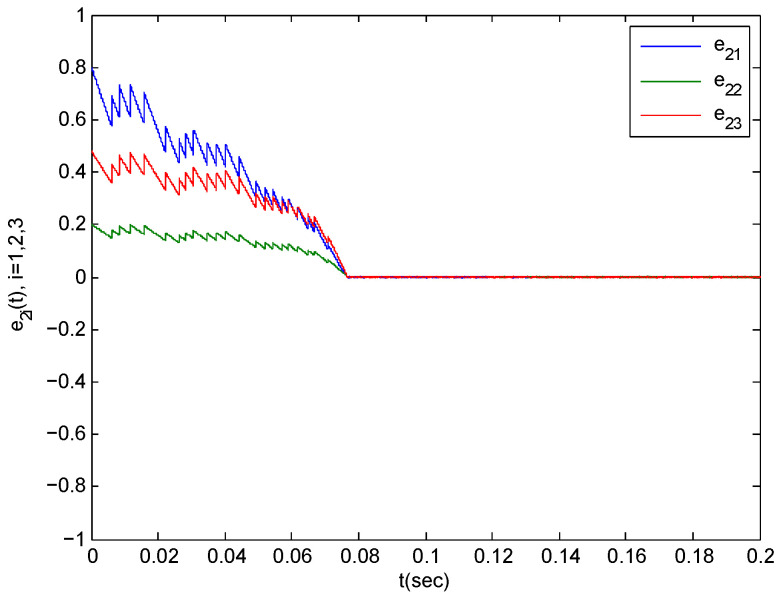
The synchronization error trajectories of complex system (45) with isolated system (46) with an initial value of $x\left(0\right)={\left(0.5,0.4,0.8\right)}^{T}$ under control.

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
