# Peer review of "Finite-Time Synchronization of Markovian Jumping Complex Networks with Non-Identical Nodes and Impulsive Effects"

_entropy, 2019, doi:10.3390/e21080779_

Round 1

Reviewer 1 Report

In this paper, the authors investigated the finite-time synchronization problem for a class ofMarkovian jumping complex networks with non-identical nodes and impulsive effects. Sufficient conditions for the Markovian jumping complex networks are presented based on the M-matrixtechnique, the stochastic analysis technique, the Lyapunov function method, and suitable comparisonsystems, to guarantee finite-time synchronization. Finally, numerical examples are provided to illustrate the theoretical results, and they demonstrate the effectiveness of our results for complexsystems. This paper is interesting, but the following minor revisions should beconsidered. 

1.A few grammar mistakes and writing errors exist in this paper. Please check it carefully. eg. Page 2, line 63 “…ih the previous literature,…”?   Page 9, line 189, “then we can obtain that…..” should be “Then we can obtain that…..”   Page 13, line 231, “chose” should be “choose” Page 13, line 238, “studied” should be “have studied”

2. The introduction is very simple, and the motivation and research background should be clarified and highlighted. Assumption 2 is too strong. 

3. I don't see the math difficulty or skill since the math derivation is standard. The authors should give a explain;

 4. The results are derived from the concise derivations and some strong conditions; however, what's the weakness of the obtained results?

 5. What is the importance of the finite-time synchronization? Please explain it in detail.

 6. The following recently papers on synchronization, impulsive controls and Lyapunov stability should be cited and discussed: Nonlinear Analysis: Real World Applications, 2011 (12) (1)93-105. Nonlinear Analysis: Hybrid Systems, 2011 (5) (1)52-77. Journal of Applied Mathematics,Volume 2012, 2012, ID 974639, 11 pages. Abstract and Applied Analysis,Volume 2012, 2012, ID 603535, 25 pages. Journal of Mathematical Biology,2018 (76)(6)1387-1419.  International Journal of Biomathematics,12 (1) (2019) 1950016. Communications on Pure and Applied Analysis, 18 (6)(2019),3337-3349. Complexity, 2018:1–10, 2018.

Author Response

      We would like to thank the associate editor and the reviewers for your affirmation and objective appraisal on our original manuscript. We have taken into account your comments seriously, have addressed thoroughly your concerns and have revised the paper according to your valuable suggestion and comments. Major changes in the revised manuscript are highlighted as follows:

1、In the revised version, we use ‘red’ text to highlight those parts that have been revised.

2、We have revised thoroughly the manuscript to refine the language usage.

3、Reviewer's comments are provided a point-by-point response in the attachment.

Reviewer 2 Report

In addition to some technical questions as raised in the attached pdf file, I have noticed that following presentation issues, e.g.,

In Abstract, “the M-matrix technique, the stochastic analysis technique, the Lyapunov function method, and suitable comparison systems” => “an M-matrix technique, stochastic analysis technique, Lyapunov function method, and suitable comparison systems”.

Line 1, Sec. I, “have improved” => “improved” due to the past time phrase “Over the past decades”. Line 2, “have” => “had”.

Page 1, Line -8, “From practical point of view” => “From a practical point of view”. Line -5, “the finite-time” => “a finite-time”

Page 2, Line 3, “they can have” => “yield”

The following papers are related:

[A]   Weiyuan Ma, Yujiang Wu, and Changpin Li, “Pinning Synchronization Between Two General Fractional Complex Dynamical Networks With External Disturbances,” IEEE/CAA Journal of Automatica Sinica, 2017, 4(2), 332-339

[B]   J. Zhu, Q. Ding, M. Spiryagin, and W. Q. Xie, “State and mode feedback control for discrete-time Markovian jump linear systems with controllable MTPM,” IEEE/CAA J. Autom. Sinica, vol. 6, no. 3, pp. 830-837, May 2019.

[C]  Yanxia Tan and Zhenkun Huang, "Synchronization of drive-response networks with delays on time scales",  IEEE/CAA Journal of Automatica Sinica ( Early Access ), DOI: 10.1109/JAS.2016.7510043, On-line, 2018.

Author Response

(The authors gave the same response as above.)
